# The Discovery of the Mode of Action of Nitisinone

**DOI:** 10.3390/metabo12100902

**Published:** 2022-09-25

**Authors:** Edward A. Lock

**Affiliations:** School of Pharmacy and Biomolecular Sciences, Liverpool John Moores University, Byrom Street, Liverpool L3 3AF, UK; edwardlock_600@hotmail.com

**Keywords:** Nitisinone, 4-hydroxyphenylpyruvate dioxygenase, tyrosinaemia type1, alkaptonuria

## Abstract

This review briefly discusses the discovery of the mode of action of the triketone herbicide, 2-(2-nitro-4-trifluormethylbenzoyl)-1,3-cyclohexanedione and its use as a drug Nitisinone for the treatment of inborn errors of tyrosine metabolism. Nitisinone is a potent reversible tight-binding inhibitor of the enzyme 4-hydroxyphenylpyruvate dioxygenase, involved in the catabolism of the amino acid tyrosine. Nitisinone is used to treat the rare disease hereditary tyrosinaemia type 1 where the last enzyme in the breakdown of tyrosine, fumarylacetoacetase is deficient. Nitisinone is also used to treat patients with alkaptonuria where the enzyme homogentisic acid oxidase is deficient. Articles in this issue discuss metabolites of tyrosine catabolism in healthy patients and those with alkaptonuria.

## 1. Introduction

### The Discovery of the Mode of Action of Nitisinone

It is important in this issue of the journal to discuss the background to the use of Nitisinone, as it is the basis for many of the papers that follow. In addition, the use of proton magnetic resonance spectroscopy in our studies on the mode of action of Nitisinone, conducted by Ian Wilson, who is a great exponent of metabolomics, provided a crucial lead in our investigation.

Nitisinone, 2-(2-nitro-4-trifluoromethylbenzoyl)-1,3-cyclohexanedione (Figure 1) was one of a series of chemicals being developed in a novel class of triketone herbicides. Triketones were shown to have broad spectrum activity on grass and broadleaf weeds and have selectivity in corn [1]. Triketone-treated plants are bleached with reduced chlorophyll and carotenoids and elevated phytoene levels. Nitisinone applied to crops caused burn as well as bleaching during testing, which stopped its development. However, structurally related triketones were developed some which were good selective herbicides [2]. The mode of action in plants was not known, and a combination of work by the plant biochemists and synthetic chemists at Zeneca Agrochemicals at the Western Research Center, California and toxicologists undertaking safety evaluation on these chemicals in California and at Zeneca Central Toxicology Laboratories in Cheshire led to the discovery of the mode of action of this chemical series.

Early toxicology studies were conducted at the Environmental Health Center Laboratories at Farmington CT, USA where they showed that Nitisinone is not acutely toxic, but following repeated exposure caused ocular toxicity. At Alderley Park, Cheshire studies in rats showed that Nitisinone caused a reversible lesion to the surface of the cornea of the eye after daily doses as low as 1 mg/kg/day for 6 weeks [3]. This finding was the main toxicological response observed, but was confounded by a species difference in response. Beagle dogs showed eye lesion at low doses of Nitisinone like the rats, which was reversible upon stopping exposure to Nitisinone. However, no corneal lesions were seen in mice, rabbits or rhesus monkeys given daily doses of 10 mg/kg/day for 90 days [4].

Scientists in California noted reports in the literature describing inhibitors of the enzyme tyrosine hydroxylase with some of the inhibitors having a structural resemblance to triketones. They analysed urine from Nitisinone-treated rats for tyrosine and its metabolites using ferric chloride and nitrosonaphthol that would detect 4-hydroxyphenylpyuvate (HPPA) and tyrosine, HPPA and 4-hydroxyphenyllactate (HPPL), respectively, the urine was positive with both reagents. They also showed that plasma from Nitisinone-treated rats had high levels of the amino acid tyrosine [1]. Parallel studies at Alderley Park, Cheshire, on urine from untreated dogs and dogs-treated with Nitisinone were examined using ^1^H-nuclear magnetic resonance spectroscopy at Zeneca Pharmaceutical by Ian Wilson. This showed the appearance of two doublet peaks at about 7 and 7.2 ppm which were not seen in untreated urine. These peaks were also seen in urine from rats treated with Nitisinone and identified as HPPA and HPPL consistent with the studies in California [5]. Amino acid analysis on plasma taken from rats dosed with Nitisinone showing eye lesions, and those with no eye lesions had selective elevation of tyrosine while other amino acids where similar to those in untreated rats [3]. This indicated an effect on tyrosine catabolism and with HPPA and HPPL appearing in urine suggested a block at the level of 4-hydroxyphenyl pyruvate dioxygenase (HPPD). The enzyme activity of adrenal tyrosine hydroxylase, liver tyrosine aminotransferase, HPPD and homogentisic acid oxidase determined in the presence of Nitisinone, showed it was a potent inhibitor of HPPD from rat liver, while not effecting other enzymes in the pathway [5] (Figure 1). Thus the mode of action of a novel series of herbicides had been discovered and the presence of HPPD in plants confirmed.

The tyrosinaemia and ocular lesions produced by inhibition of HPPD in rats is similar to that reported when feeding rats, a high concentration of tyrosine in their diet [6], leading us to conclude that the corneal lesions produced by Nitisinone are a result of a marked and sustained ocular tyrosinaemia.

## 2. Results and Conclusion

### Use of Nitisinone as a Drug to Treat Rare Hereditary Disorders of Tyrosine Catabolism

Inborn errors in tyrosine metabolism occur, but are generally rare. A metabolic deficiency of tyrosine aminotransferase (Figure 1), Richner-Hanhart syndrome leads to a marked tyrosinaemia which can result in corneal opacities if the dietary intake of phenylalanine and tyrosine is not restricted [7]. A small number of cases of HPPD deficiency have been reported [8,9] called tyrosinaemia type III (Figure 1). Tyrosinaemia Type 1 is where fumarylacetoacetase, the last enzyme in the pathway is deficient resulting in acute tissue injury with children usually dying in the first two years of life.

In collaboration with Professor Lindstedt and his colleague Dr Elisabeth Holme, at the University of Gothenburg, Sweden, five patients with hereditary tyrosinaemia type 1 were treated with Nitisinone provided by Zeneca Agrochemicals and the first serious ill patient made a remarkable recovery. This was reported in The Lancet [10] for the background to using the herbicide as a drug see [11]. The chemical was patented for clinical use world-wide by Zeneca Pharmaceuticals and the product sub-licensed to Swedish Orphan AB, now called Swedish Orphan Biovitrum AB for the treatment of this metabolic disorder. With the drug available world-wide and authorised by the United States Food and Drug administration in 2002 and the European Medicines Agency in 2005, the number of reported cases has increased. There is now a survival rate of greater than 90% in children treated soon after birth. Side-effects in the eye has been reported in a small number of cases, which is reversible if treatment is stopped, but it is recommended to keep plasma tyrosine concentration < 500 nmol/mL and use a tyrosine and phenylalanine free diet. There are a number of reports of lower cognitive function and IQ as well as schooling and behavioural problems in children on Nitisinone [12,13,14,15] which may be due to the high circulating tyrosine concentration in the brain as a result of Nitisinone treatment, rather than the drug itself. What is clear is that if treatment is started in the first 28 days of life then hepatic disease is prevented [16]. Although Nitisinone has become the standard of care for hereditary tyrosinaemia type 1, long-term follow-up data on safety is still needed [17].

Nitisinone is also being used to treat other disorders including oculocutaneous albinism [18] and alkaptonuria (Figure 1) [19,20]. It is this last inborn error in metabolism that is the focus of research in this issue of the journal, with particular emphasis on metabolomic approaches to understanding the consequences of Nitisinone treatment. Unlike tyrosinaemia type 1 which causes acute and chronic tissue injury alkaptonuria cause a chronic disease of the joints called ochronosis [20,21]. Treatment of patients with Nitisinone at 2 mg/day, inhibited ochronosis and thereby slowed the progression of tissue damage, by effectively inhibiting the production of homogentisic acid. It should be noted that the dose of Nitisinone is considerably lower than that used for tyrosinaemia type 1. Dietary restriction is required to keep plasma tyrosine < 500 nmol/mL, but many patients were less compliant with the low protein diet leading to some tyrosine values > 900 nmol/mL three patients developing a keratopathy, one which was silent, which resolved with a lower protein diet enabling treatment with Nitisinone to continue. However, half-yearly slit lamp examination of eyes is recommended for patients with tyrosine levels > 900 nmol/mL. A recent paper has reported the incidence of nuclear and cortical opacities in a large cohort of patient with alkaptonuria who have been treated with Nitisinone over a 5-year period and compared with a small number of alkaptonuria patients not on Nitisinone [22]. In the UK the prevalence of cataracts causing visual impairment in those over 65 is around 30%. In contrast, the overall prevalence of cataract in the alkaptonuria patient cohort with a mean age of 44 was 76%. The main factor between the two groups is Nitisinone-induced tyrosinaemia in the plasma and eyes, with homogentisic acid/pigmentation in the eye, but dietary factors cannot be excluded, further studies are need to understand this effect in more detail. The incidence of cataract in tyrosinaemia type 1 patients on Nitisinone is 1%, but the patients are much younger.

## Figures and Tables

**Figure 1 metabolites-12-00902-f001:**
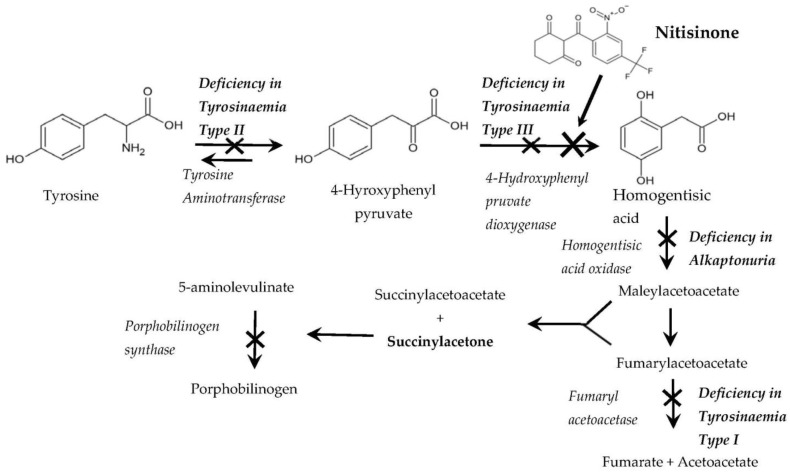
The metabolic pathway for tyrosine catabolism in humans and experimental animals, showing the enzymes involved, the known sites of inborn errors in metabolism and the site of action of the drug Nitisinone used to treat tyrosinaemia type 1 and alkaptonuria.

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
