# Peer review of "The Discovery of the Mode of Action of Nitisinone"

_metabolites, 2022, doi:10.3390/metabo12100902_

Round 1

Reviewer 1 Report

This is an excellent mini-review regarding the discovery and mode of action of nitisinone. It describes the mode of action and clinical application of nitisinone which is of interest to the clinician.

The paper is well written and easy to follow.

Though there is no novel finding in the manuscript it is a nice introduction for a series of manuscripts on nitisinone.

There are no ethical issues.

Author Response

Comments to reviews 1 and 2

  • Thank you both for your comments on the manuscript. I have added some more information with regard to animal toxicity highlighted in red
  • Discussed in more detail the side effects in patients which are mainly but not exclusively related to the eye as a consequence of prolonged and sustained tyrosinaemia. The need for dietary control to ideally keep tyrosine levels < 500nmol/ml. also highlighted in red
  • Added more detail on the response of AKU patients to Nitisinone with the 3 cases of corneal injury and finished on the latest finds of a small increase in cataract formation in AKU but not in HT-1 patients, as much younger, on Nitisinone.
  • Many thanks
  • Ted Lock

Reviewer 2 Report

Journal: Metabolites (ISSN 2218-1989)

Manuscript ID: metabolites-1918780

The discovery of the mode of action of Nitisinone

Edward Lock

Abstract: This review briefly discusses the discovery of the mode of action of the triketone herbicide, 2-(2-nitro-4-trifluormethylbenzoyl)-1,3-cyclohexanedione and its use as a drug Nitisinone for the treatment of inborn errors of tyrosine metabolism. Nitisinone is a potent reversible tight-binding inhibitor of the enzyme 4-hydroxyphenylpyruvate dioxygenase, involved in the catabolism of the amino acid tyrosine. Nitisinone is used to treat the rare disease hereditary tyrosinaemia type 1 where the last enzyme in the breakdown of tyrosine, fumarylacetoacetase is deficient. Nitisinone is also used to treat patients with alkaptonuria where the enzyme homogentisic acid oxidase is deficient. Articles in this issue discuss metabolites of tyrosine catabolism in healthy patients and those with alkaptonuria.

Comments:

My detailed comments are as follows:

It is a topic of interest for researchers in the related area.

I consider that the revision does not need major improvements before being accepted for publication.

The correct title of the review makes clear the author's intentions in developing the manuscript. It is important in this issue of the journal to discuss and review the background of the use of nitisinone, since it is the basis of many of the articles that follow (with this manuscript a real historical reference is made, of why nitisinone is used in present). In addition, the use of proton magnetic resonance spectroscopy (as the author comments...) in different studies on the mode of action of nitisinone, provided a crucial clue in the investigations developed in the last decades. Furthermore, as the author of the manuscript says, although nitisinone has become the standard treatment for hereditary tyrosinemia type 1, long-term follow-up data on safety are still needed.

I have not detected excessive citations of the author in the manuscript. Also the total number of citations is low. I think that perhaps some additional references to the use of nitisinone in animal models (toxicity assessment...) and perhaps in human populations could be included.

I consider, once again as I have mentioned before, that minimal interventions are necessary to increase the (already good) quality of the review.

Author Response

(The authors gave the same response as above.)
